# Cat caregivers' perceptions, motivations, and behaviours for feeding treats: A cross sectional study

**Shelby A. Nielson**[1]*, **Deep K. Khosa**[1], **Adronie Verbrugghe**[2], **Katie M. Clow**[1]

**1** Department of Population Medicine, Ontario Veterinary College, University of Guelph, Guelph, Ontario, Canada, **2** Department of Clinical Studies, Ontario Veterinary College, University of Guelph, Guelph, Ontario, Canada

* snielson@uoguelph.ca

## Abstract

There is an abundance of research focusing on the nutritional needs of the cat, though aspects surrounding treat feeding have received far less attention. Feeding practices have the potential to cause nutrient imbalances and adverse health outcomes, including obesity. The objective of this study was to identify and describe the perceptions, motivations, and behaviours surrounding treats, and factors that influence treat feeding by cat caregivers. A 56-question online survey was disseminated to cat caregivers (n = 337) predominantly from Canada and the USA to collect data regarding caregiver and cat demographics, the pet-caregiver relationship, perceptions surrounding treats, and feeding practices and behaviours. Descriptive statistics, chi-square tests, Kruskal-Wallis one-way ANOVA, Wilcoxon signed-rank tests, and multivariable logistic regression models were used to analyze the survey data. Caregivers had varying interpretations of the term 'treat' and how treats relate to the primary diet, and these perceptions appeared to influence the quantity of treats provided. Aspects relating to the human-animal bond were highlighted as an important factor in decisions and behaviours surrounding treat feeding in our results. Though the majority (224/337, 66%) of respondents indicated they monitor their pet's treat intake, using an eyeball estimate was the most frequent (139/337, 41%) method reported to measure treats. Multivariable logistic regression results revealed feeding jerky, bones, dental treats, and table scraps in select frequencies were predictive of caregivers perceiving their cat as overweight/obese. Results provide valuable new insights to cat caregiver feeding practices and perceptions of treats and can be used to inform veterinary nutrition support to caregivers. More research is warranted to further our understanding and ensure that cats receive optimal nutrition and care.

## Introduction

Domestic cats are among the most common companion animals and play an important role in the lives of humans in various ways, through provision of therapeutic support and

---

**Data Availability Statement:** The data has been uploaded to a public repository. DOI: 10.5683/SP3/EEAXSO Reference: Nielson, Shelby, 2023, "Supplementary Data: Cat caregivers' perceptions, motivations, and behaviours for feeding treats: a

cross sectional study", https://doi.org/10.5683/SP3/EEAXSO, Borealis, V1, UNF:6:bY9XKkmnQgG4QdLd5am8fg== [fileUNF]

**Funding:** The author(s) received no specific funding for this work.

**Competing interests:** A.V. is the Royal Canin Veterinary Diets Endowed Chair in Canine and Feline Clinical Nutrition and declares that they serve on industry advisory boards and received honoraria and research funding from various pet food manufacturers and ingredient suppliers.

companionship [1, 2]. In Canada and the U.S., approximately 35% of households have at least one cat [3–5], with cats outnumbering dogs in Canadian households [6]. In 2020, an estimated 8.1 million cats were considered households pets in Canada [6].

Feline obesity has become a prevalent issue and is recognized as one of the leading health problems in domestic cats in developed countries, second only to dental disease [7–9], and is the most common multifactorial nutritional disorder in cats [10, 11]. Companion animals are generally considered overweight or obese when excess body fat exceeds 20% and 30%, respectively [12, 13]. Global estimates of overweight and obesity in domestic cats are reported to range from 27–63% [7, 14–16], and from 35–41% in North America specifically [17, 18]. Obesity is associated with numerous comorbidities in companion animals including diabetes mellitus, neoplasia, urinary disorders, and oral disease, along with reduced quality of life and longevity [13].

While a broad range of factors, such as breed, age, sex and spay/neutering [17, 19, 20] have been identified as potential risk factors for the development of obesity in cats, feeding practices have been demonstrated as a consistent predictor. Previous studies have identified feeding practices such as ad libitum feeding [14, 21, 22], feeding meals two or three times per day [23], and feeding dry food as the only or majority food type at one and two years of age [24, 25] to be associated with higher odds of feline obesity. Treat feeding has also been shown to contribute to obesity in domestic cats, with evidence suggesting that supplementary feeding practices, such as the provision of kitchen scraps and treats, increase the risk of obesity [14, 22, 25].

Despite most companion animal caregivers providing treats [26–30], including cat caregivers [31], research focusing on caregiver feeding practices related to treats remain underexplored. White et al., [32] investigated owner perceptions and motivations for treat giving with dog owners, and Morelli et al., [33] explored dog owner attitudes towards treats, though to date, research has yet to focus on treat feeding with cats exclusively. Treats are often provided as a means of positive reinforcement during training [32, 33], and previous literature has demonstrated that caregivers often associate the provision of food with showing love or affection [32, 34]. Rowe et al., [25] revealed that owners who provided treats as a reward were more likely to have an overweight or obese cat, but current literature has not addressed other reasons for which caregivers may be motivated to provide treats to their cats. Moreover, research has yet to explore aspects relating to cat caregiver behaviours and decisions surrounding treat selection and provision, and the perceived role of treats in relation to the primary diet. To address these gaps in the literature, this exploratory study aimed to identify cat caregivers' perceptions and motivations around when, how and why they choose to feed their cats treats. Additionally, this study aimed to describe cat caregivers' perceptions of the term 'treat' and how they consider treats in relation to the cat's primary diet. Synonymous with the title 'cat owner' that is often presented, the commitment in caring for the physical and emotional well-being of a cat has been represented through use of the term 'caregiver' in this study. An improved understanding of cat caregiver perceptions and feeding practices associated with treats is an essential step to promote weight management and improve feline health and wellbeing.

## Methods

Pet caregivers who self-declared that they feed treats were recruited to complete an online questionnaire from September 20, 2021 to October 31, 2021. Inclusion criteria for this cross-sectional study included being a primary caregiver of at least one cat or dog; the responses from cat caregivers were included in these analyses. This study received University of Guelph's Research Ethics Board approval (REB #21-06-014).

## Questionnaire design

A review of existing literature and discussions amongst the research team informed the development of the questionnaire. The questionnaire included written informed consent followed by five main sections with questions about (1) caregiver demographics; (2) cat characteristics; (3) pet-caregiver relationship (4) perceptions surrounding treats; and (5) feeding practices and behaviours. Pilot testing for survey length, question clarity, and comprehensiveness was completed by a sample of 15 pet caregivers from the research team's home institution. Feedback from the piloting process was used to further refine question wording and establish content validity. Questions in the final version of the survey were presented in multiple-choice, 5-point Likert scale, and sliding scale formats. The final questionnaire included 56 questions. Question item reliability was calculated at Cronbach's alpha = 0.88, yielding very good internal consistency [35]. For select multiple-choice questions, respondents had the opportunity to select more than one response and further specify or elaborate on their response using an "other" category with optional free-text. Most questions also presented the option "prefer not to answer" for respondents. In the case that respondents identified as the caregiver of multiple pets, they were asked to complete the survey with respect to one animal only. The term "other people in the household" was used inclusively to include any individuals living in the household and potentially contributing to the pet's care, including family members. To investigate the body condition score (BCS) of cats, caregivers were asked, "which diagram best illustrates the body condition of your pet" where images resembling the 5-point BCS chart from the American Animal Hospital Association© (1/very thin, 2/underweight, 3/normal/ideal, 4/overweight, 5/obese) [36]. This scoring system is commonly referenced by veterinarians [37] and hence, caregivers were likely to understand and be familiar with it. Diagrams were presented in a random order and did not include category labels or written descriptions (e.g., "underweight", "obese") to avoid bias. Questions pertaining to the present study can be found in supplementary material (S1 Appendix).

## Questionnaire distribution

The questionnaire was distributed online using the survey software Qualtrics© (2017, Provo, UT). Recruitment was completed using an infographic posted to a variety of social media platforms including Facebook, Instagram, Twitter, and LinkedIn, including from the Ontario Veterinary College social media accounts. Posts had the ability to be shared widely to reach a broad audience using a snowball sampling approach; the aim was for respondents to represent a convenience sample of pet caregivers from the general public. The questionnaire was accessible for 6 weeks from September to November 2021, and was available in the English language only. The questionnaire was anonymous; responses were not linked to any personal identifiers. Participants were required to be at least 18 years of age or older, identify as the primary guardian of at least one cat, and feed treats to their cat. In the case that respondents cared for more than one cat, they were asked to place their cat's names in alphabetical order and complete the survey with respect to the animal whose name starts with the letter closest to 'A'. Upon submission of the questionnaire, respondents were provided with a separate, external link to enter an optional incentive prize draw for 1 of 23 Amazon gift cards ranging in value from $20–100. All personal data collected for the prize draw were stored separately from the main survey data and were permanently deleted after the winning participants were contacted.

## Data analysis

Data from the questionnaire was exported into a Microsoft Excel spreadsheet and uploaded to the statistical software package STATA 15.1© [38] for quantitative analyses. Distributions were

verified for all variables. Country of residence were categorized to: Canada, United States of America (USA), and other (neither Canada nor USA). The frequency variables methods of delivery for primary diet and treats, and reasons treats are fed, were dichotomized where categories 1, 2 and 3 (never, rarely, sometimes) were combined as 0 / "never, rarely", and categories 4 and 5 (very often, always) were combined as 1 / "often". Factors likely to influence what treats caregivers feed to their pets was dichotomized where categories 1, 2 and 3 (very unlikely, unlikely, neither likely/unlikely) were combined as 0 / "unlikely", and categories 4 and 5 (likely, very likely) were combined as 1 / "likely". The variable how often different treats are fed was dichotomized for some analyses where categories 1, 2 and 3 (never, monthly, weekly) were combined as 0 "never/infrequently", and categories 4 and 5 (a few times a week, daily) were combined as 1 / "often" (0 = never/infrequently, 1 = often*)*. Descriptive statistics were calculated for each survey item, including frequency distributions, and mean (standard deviation) and median (range) responses depending on data distribution.

To investigate associations between cat caregiver age (categorical variable) and the variables (1) type of relationship with cat (categorical variable), (2) factors relating to feeding the primary diet (dichotomous variables), and (3) factors relating to feeding treats (dichotomous variables), chi-square tests of associations were used. Chi-square tests of associations were similarly used to explore associations between the dichotomous variables method of measurement used for the primary diet versus treats, how caregiver decides on amount to feed for the primary diet versus treats, and the method of delivery for the primary diet versus treats. Preliminary exploration of caregiver-reported BCS on a 5-point scale (categorical variable) and types of treats fed frequently to cats (dichotomous variable) were also done using chi-square tests. All significant ($\alpha$<0.05) association were subsequently investigated using Bonferroni multiple comparisons test to determine directionality. In the case of few observations (<10) per cell for all chi-square tests of associations, Fisher's exact tests were used.

The Kruskal-Wallis one-way ANOVA was used to explore the relationship between 1) the variable percent of diet composed of treats (based on estimated quantity; continuous variable) by caregivers' perceptions of treats in relation to the normal diet (categorical variable), and 2) the variable percent of diet composed of treats (based on estimated quantity; continuous variable) by caregivers' reported level of attachment to their pet (categorical variable). The post-hoc Dunn's test was then used to investigate significant ($\alpha$ <0.05) differences in estimated percent of diet composed of treats by caregivers' perceptions of treats in relation to the normal diet, and differences in estimated percent of diet composed of treats by caregivers' reported level of attachment to their pet.

To explore the association between pet's perceived feelings upon receiving a treat as reported by caregiver (continuous variable) and caregiver's perceived feelings upon providing a treat to pet (continuous variable), a Wilcoxon signed-rank test was conducted ($\alpha$<0.05). The post-hoc Dunn's test was then used to investigate significant ($\alpha$<0.05) differences between perceived feelings.

Next, two logistic regression models were built to explore the associations between (1) whether caregivers monitor their cat's treat intake (yes/no outcome), and method of which they measure their cat's treats (explanatory dichotomous yes/no variables), and (2) caregiver-reported BCS as overweight/obese (dichotomous outcome) and frequency of feeding different types of treats (explanatory dichotomous yes/no variables) to their cat.

For the first logistic regression model, the 3-level categorical variable if caregivers monitor their pet's treat intake was re-coded to a 2-level variable: 0/ "no" (original values 1/no, 2/sometimes) and 1/ "yes" (original value 3/yes). The categorical variables caregiver age, gender, education, cat age, sex, and caregiver-reported BCS were also included in this analysis due to their suspected role as confounding variables on the association of interest.

For the second logistic regression model, the 5-level BCS was re-coded to a 2-level variable: 0/ "not overweight/obese" (original values 1/very thin, 2/underweight, 3/normal/ideal, 6/ unsure) and 1/ "overweight/obese" (original values 4/overweight, 5/obese). The following categorical variables were included in this analysis due to their suspected role as confounding variables on the association of interest: caregiver age, gender, education, cat age, and sex.

For both predictive models, univariable analysis was first conducted using logistic regression models to screen the explanatory variables, including the suspected confounding variables, with the outcome variables. Variables with a liberal significance level ($\alpha \leq 0.2$) were considered for inclusion in the multivariable analysis. Spearman's correlation analysis ($|r| > 0.7$) was then performed to determine whether there were any pairwise correlations between the independent variables. The multivariable logistic regression models were then built using manual backward-selection whereby potential explanatory variables were removed from the multivariable model in order of least significance. A likelihood ratio test ($\alpha < 0.05$) was performed prior to removing any categorical variables that did not appear statistically significant. Independent variables that did not have a statistically significant main effect were further evaluated for confounding effects before they were removed from the model. Confounding was tested by determining the absolute difference between the coefficient of the variable(s) thought to be impacted by the potential confounder in the crude and adjusted model using the equation: |(ln crude–ln adjusted)|/ln crude. If the difference in the coefficients between the crude and adjusted models was $\geq 20\%$, the variable was considered to be a confounding variable and was retained in the model. For each model, multicollinearity was assessed using the uncentered variance inflation factor (VIF<10). Further, area under the receiver operating characteristic curve (AUROC) were used to measure the model's ability to discriminate between outcomes of interest [39, 40]. Both models yielded $0.8 \leq$ AUROC $< 0.9$, indicating excellent discrimination [39, 40]. The goodness-of-fit of the models were then assessed based on Pearson and Hosmer-Lemeshow tests ($\alpha < 0.05$).

## Results

### Caregiver and household demographics

A total of 1,271 responses had been recorded when the questionnaire closed. Questionnaires completed with regards to a dog (716 responses) and incomplete questionnaires (218 responses) were excluded, which left a total of 337 questionnaires for analysis. Survey participants primarily comprised of cat caregivers aged 26–34 (134/337, 40%), and most (270/337, 80%) respondents identified as a woman. Canadian residents (270/337, 82%) were more represented than were USA residents (54/337, 16%) (Table 1).

### Cat characteristics

Most caregivers reported their cats as neutered males (143/337, 42%) or spayed females (109/ 337, 32%), and half (168/337, 50%) of the cats described were between 1–3 years of age (Table 2). Respondents most commonly rated their cat as 3/ "normal/ideal" (148/337, 44%) based on images from a 5-point body condition score (BCS) chart; 14% (46/337) and 15% (52/ 337) of respondents rated their cat as 4/ "overweight" and 5/ "obese", respectively.

### Pet-caregiver relationship

Most (170/337, 51%) caregivers described their cat as part of their family, followed by like a child (117/337, 35%) (Table 2). Though there were significant (p<0.025) differences in these types of relationships by caregiver age group, with higher proportions of caregivers in younger

**Table 1. Cat caregiver demographics (n = 337).**

| Variable | | n | % |
|---|---|---|---|
| Age (years) | 18–25 | 96 | 28.5 |
| | 26–35 | 134 | 39.8 |
| | 36–45 | 47 | 14.0 |
| | 46–59 | 39 | 11.6 |
| | 60+ | 20 | 5.9 |
| | Prefer not to answer | 1 | 0.3 |
| Gender | Woman | 270 | 80.1 |
| | Man | 61 | 18.1 |
| | Non-binary | 5 | 1.5 |
| | Prefer to self-describe | 1 | 0.3 |
| Residence | Canada | 272 | 82.0 |
| | United States of America | 54 | 16.2 |
| | Other (neither Canada nor USA) | 6 | 1.8 |
| Education | Some high school | 7 | 2.1 |
| | High school | 28 | 8.3 |
| | Apprenticeship training and trades | 3 | 0.9 |
| | College | 69 | 20.5 |
| | Bachelor's degree | 121 | 35.9 |
| | Master's degree | 61 | 18.1 |
| | Professional degree | 34 | 10.1 |
| | Ph.D. or higher | 13 | 3.9 |
| | Prefer not to answer | 1 | 0.3 |
| Career in companion animal care | Yes | 70 | 20.8 |
| | No | 266 | 78.9 |
| | Prefer not to answer | 1 | 0.3 |
| Number of other people living in household | None | 49 | 14.5 |
| | 1 | 101 | 30.0 |
| | 2–4 | 135 | 40.1 |
| | More than 4 | 50 | 14.8 |
| | Prefer not to answer | 2 | 0.6 |
| Number of other people living in household that contribute to pet's care | None | 141 | 41.8 |
| | 1 | 126 | 37.4 |
| | 2–4 | 61 | 18.1 |
| | More than 4 | 9 | 2.7 |
| | Prefer not to answer | 0 | 0.0 |
| Total number of cats in household | 1 | 188 | 55.8 |
| | 2 | 101 | 30.0 |
| | 3+ | 48 | 14.2 |
| | Prefer not to answer | 0 | 0.0 |

age groups describing family and child-like relationships compared to caregivers in older age categories. There were no significant differences among other types of relationships by caregiver age groups. When asked to rate their level of attachment with their cat on a scale of 1–10 (0 = not attached, 10 = very attached), caregivers indicated an average level of attachment of 8.70 (median = 9, range = 3–10). Over 80% (282/337, 84%) of caregivers rated their level of attachment as $\leq$ 8/10, with 40% (136/337) indicating their level of attachment as 10/10.

**Table 2. Cat demographics (n = 337).**

| Variable | | n | % |
|---|---|---|---|
| Sex | Male intact | 56 | 16.6 |
| | Male neutered | 143 | 42.4 |
| | Female intact | 29 | 8.6 |
| | Female spayed | 109 | 32.3 |
| Age | <1 year | 21 | 6.2 |
| | 1–3 years | 168 | 49.9 |
| | 4–6 years | 63 | 18.7 |
| | 7+ years | 85 | 25.2 |
| BCS (5-point) | Very thin (1) | 47 | 14.0 |
| | Underweight (2) | 38 | 11.3 |
| | Normal/ideal (3) | 148 | 43.9 |
| | Overweight (4) | 46 | 13.7 |
| | Obese (5) | 52 | 15.4 |
| | Unsure | 6 | 1.8 |
| Existing health concern | Yes | 71 | 21.1 |
| | No | 240 | 71.2 |
| | Unsure | 26 | 7.7 |

## Perceptions surrounding treats

While most (147/337, 44%) cat caregivers considered the term 'treat' as "anything I give my pet that they enjoy", more than a quarter of respondents also considered "any type of food I give my pet that they enjoy" (94/337, 28%) and "products sold exclusively as treats for pets" (88/337, 26%) (Table 3). Caregivers also revealed varied considerations to how they perceive treats in relation to their cat's normal diet, with 42% (140/337) of respondents considering treats part of their pet's normal diet, 30% (102/337) considering treats an extra to their pet's normal diet, and 28% (93/337) considering some treats part of their pet's diet, while others are extra (Table 3). When asked to select which food items caregivers considered to be 'treats' for pets, commercial pet treats were indicated by most (244/337, 72%) caregivers (Table 3). Nearly half of all respondents considered pet food given outside of regular mealtimes (166/337, 49%), human food prepared specifically for their pet (162/337, 48%), and commercial dental treats (158/337, 47%) as treats for their cat (Table 3).

Caregivers rated their perceptions of how their cat feels when they receive a treat (median = 5, range = 3–5), and how they themselves feel when they provide a treat to their cat (median = 5, range = 3–5), very highly (0 = very sad, 5 = very happy). Though caregivers' rating of how they themselves feel upon providing a treat to their cat was significantly higher (Z = -5.80, p<0.001) indicating they feel very happy when feeding their cat treats.

## Feeding decisions and behaviours

**Primary diet: Type and frequency of feeding.** Dry food/kibble (256/337, 76%) was the most common type of diet fed to cats as reported by caregivers, followed by canned/wet food (243/337, 72%). Both therapeutic/prescription diets (45/337) and home cooked diets (43/337) were fed by 13% of respondents, followed by commercial raw diets (15/337) and homemade raw diets (13/337) which were fed by 4% of respondents. Four (1%) caregivers reported feeding an 'other' diet type. Respondents could select multiple diet types to represent their cat's diet most accurately. Most (124/337, 37%) cat caregivers reported feeding their cat a specific

Table 3. Reported perceptions of treats by cat caregivers (n = 337).

| Variable | | n | % |
|---|---|---|---|
| **How caregivers define the term 'treat'** | | | |
| | Anything I give my pet that they enjoy | 147 | 43.6 |
| | Any type of food I give my pet that they enjoy | 94 | 27.9 |
| | Products sold exclusively as treats for pets | 88 | 26.1 |
| | Other | 8 | 2.4 |
| **Caregivers' perceptions of treats in relation to normal diet** | | | |
| | Treats are part of pet's normal diet | 140 | 41.5 |
| | Treats are an additional extra to pet's normal diet | 102 | 30.3 |
| | Some treats are part of pet's diet, while others are extra | 93 | 27.6 |
| | Unsure | 2 | 0.6 |
| **Food items considered to be 'treats' for pet[*]** | | | |
| | Commercial pet treats | 244 | 72.4 |
| | Pet food given outside of regular mealtimes | 166 | 49.3 |
| | Human food prepared specifically for pet | 162 | 48.1 |
| | Commercial dental treats | 158 | 46.9 |
| | Food used to disguise medication | 81 | 24.0 |
| | Natural chews | 80 | 23.7 |
| | Table scraps | 78 | 23.2 |
| | Bones | 53 | 15.7 |
| | Fast food | 15 | 4.5 |
| | Other | 8 | 2.4 |

*Respondents could select more than one response option

amount twice daily, followed by 26% (88/337) who always have food available for their cat (free feed), and 21% (71/337) who feed a specific amount three times daily. Only 10% (35/337) of respondents reported feeding their cat a specific amount once daily, and 6% (19/337) fed by an 'other' frequency.

**Method of measurement and delivery for primary diet and treats.** The majority (487/ 337, 68%) of cat caregivers indicated that they use a measuring cup/scoop to measure their cat's primary diet, and use an eyeball estimate (139/337, 41%) to measure their cat's treats (Fig 1). Fifty-three (16%) caregivers reported using an 'other' method of measurement for their cat's treats (Fig 1); respondents commonly described this to involve counting the number of treats in the optional free-text box for this question. Most (238/337, 71%) caregivers frequently (very often/always) use a traditional food bowl to deliver their cat's primary diet, compared to 27% (92/337) caregivers who frequently use a traditional food bowl to deliver their cat's treats (Fig 1). There were significant (p<0.003) differences between the use of a traditional food bowl to deliver the primary diet by caregiver age group (Table 3). In general, a greater proportion of caregivers in older age groups reported frequently using a traditional food bowl to deliver the primary diet. Hand feeding was the most reported (222/337, 66%) frequent method of delivery for treats (Fig 1), and there were significant (p<0.001) differences in the frequent delivery of treats using hand feeding and interactive puzzle/slow feeder/food dispense balls between caregiver age groups (Table 4). A greater proportion of caregivers in younger age groups reported frequent delivery of treats using interactive puzzle/slow feeder/food dispense balls, while greater proportions of caregivers in the youngest (18–25) and older age groups (46–59, 60+) reported frequently feeding treats using hand feeding.

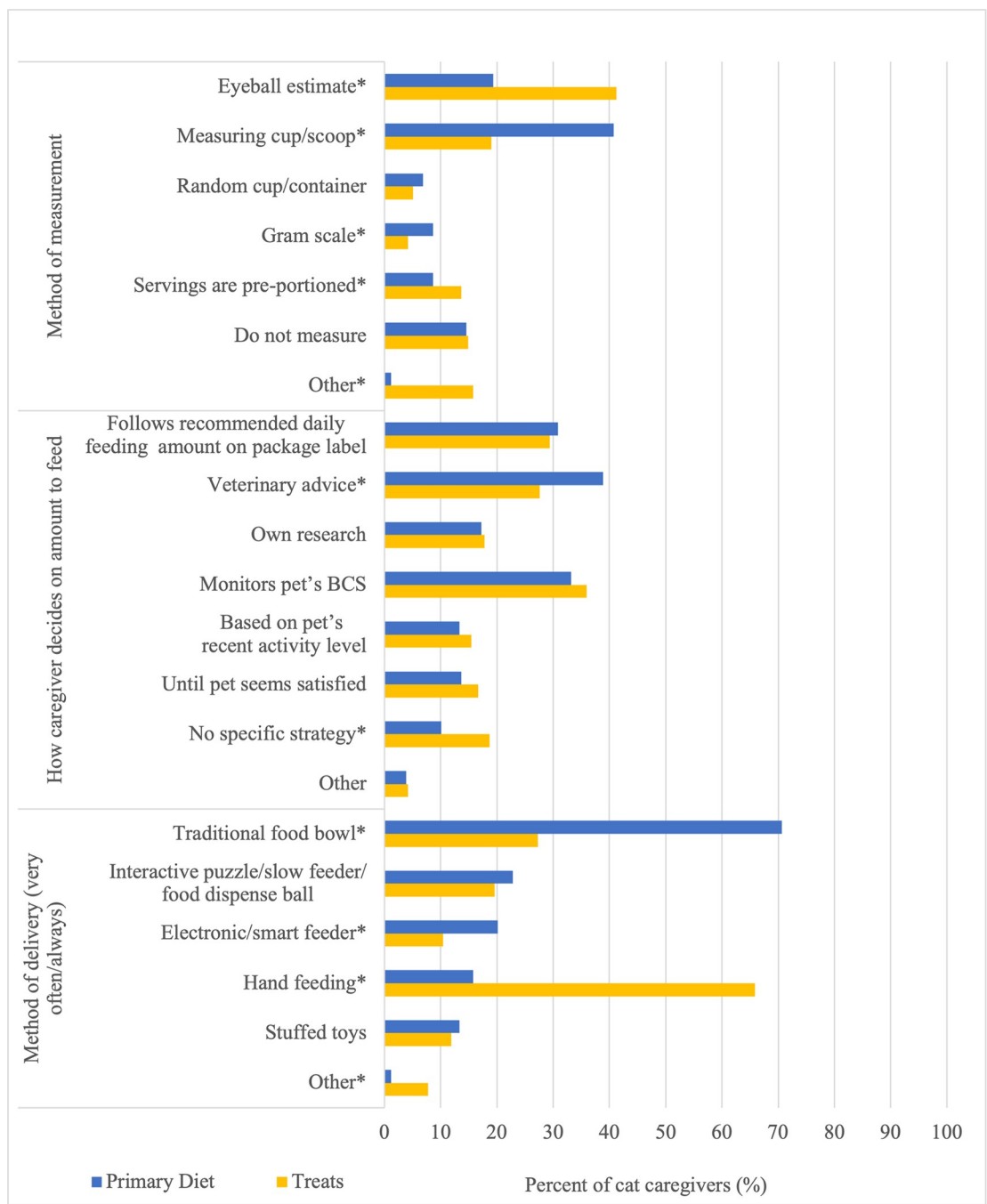

**Fig 1. Reported delivery methods, measurements, and decisions in relation to feeding the primary diet and treats by cat caregivers (n = 337).** $p < 0.05^*$ derived from two-sample test of equality for proportions for primary diet and treats. Respondents could select more than one response option for these questions.

**Method of deciding on amount to feed for primary diet and treats.** Overall, cat caregivers expressed a broad range in responses on how they decide on the amount to feed their cat for the primary diet and treats. For the primary diet, 39% (130/337) reported use of veterinary advice, while monitoring their pet's BCS (112/337, 33%) and following recommended daily feeding amounts on the food package label (104/337, 31%) were also commonly reported

**Table 4. Effect of caregiver age against type of relationship with cat, and factors relating to feeding.**

| Variable | Total (n = 337) | Caregiver age | | | | | P-Chi² | P-value |
|---|---|---|---|---|---|---|---|---|
| | | 18–25 (n = 96) | 26–35 (n = 134) | 36–45 (n = 47) | 46–59 (n = 39) | 60+ (n = 20) | | |
| **Type of relationship with cat** | | | | | | | | |
| Like a child | 117 | 38 | 53 | 19 | 6 | 1 | 17.26 | **0.001**[a] |
| Part of my family | 170 | 42 | 65 | 22 | 26 | 15 | 11.10 | **0.025** |
| Teammate/partner | 9 | 2 | 4 | 2 | 1 | 0 | 1.18 | 0.954[a] |
| Companion | 36 | 13 | 11 | 3 | 5 | 4 | 4.59 | 0.299[a] |
| Acquaintance | 3 | 1 | 1 | 0 | 1 | 0 | 1.89 | 0.756[a] |
| Prefer not to answer | 1 | 0 | 0 | 1 | 0 | 0 | 6.17 | 0.315[a] |
| **Factors relating to feeding the primary diet:** | | | | | | | | |
| **Method of deciding on amount to feed**[*] | | | | | | | | |
| Follows recommended daily feeding amount on package label | 104 | 34 | 41 | 13 | 12 | 4 | 2.26 | 0.717[a] |
| Veterinary advice | 130 | 39 | 52 | 24 | 11 | 4 | 7.94 | 0.097[a] |
| Own research | 58 | 19 | 24 | 7 | 5 | 3 | 1.26 | 0.908[a] |
| Monitors pet's BCS | 112 | 34 | 40 | 17 | 14 | 7 | 1.23 | 0.857[a] |
| Based on pet's recent activity level | 45 | 12 | 16 | 8 | 6 | 3 | 1.02 | 0.865[a] |
| Until pet seems satisfied | 46 | 10 | 26 | 3 | 5 | 2 | 6.95 | 0.162[a] |
| No specific strategy | 34 | 13 | 11 | 1 | 5 | 4 | 7.53 | 0.076[a] |
| Other | 13 | 4 | 3 | 0 | 3 | 3 | 11.07 | **0.029**[a] |
| **Method of delivery (very often/always)**[*] | | | | | | | | |
| Traditional food bowl | 238 | 69 | 91 | 26 | 33 | 19 | 15.32 | **0.003** |
| Interactive puzzle/slow feeder/food dispense ball | 77 | 24 | 40 | 7 | 5 | 1 | 11.48 | **0.021**[a] |
| Electronic/smart feeder | 68 | 26 | 29 | 8 | 4 | 1 | 8.54 | 0.077[a] |
| Hand feeding | 53 | 21 | 21 | 6 | 3 | 2 | 5.43 | 0.296[a] |
| Stuffed toys | 45 | 13 | 24 | 6 | 2 | 0 | 7.76 | 0.096[a] |
| Other | 16 | 3 | 8 | 2 | 3 | 0 | 2.76 | 0.674[a] |
| **Factors relating to feeding treats:** | | | | | | | | |
| **Factors that influence what treats caregiver feeds pet (likely/very likely)**[*] | | | | | | | | |
| Main ingredient | 245 | 74 | 95 | 32 | 31 | 13 | 3.16 | 0.523 |
| Complete ingredient composition | 192 | 59 | 75 | 28 | 21 | 9 | 2.30 | 0.680[a] |
| Shape | 96 | 30 | 38 | 14 | 12 | 2 | 3.85 | 0.406[a] |
| Size | 200 | 64 | 68 | 30 | 25 | 13 | 7.27 | 0.128 |
| Health claims | 202 | 63 | 78 | 29 | 25 | 7 | 6.99 | 0.146[a] |
| Moisture content/texture | 163 | 51 | 65 | 21 | 19 | 7 | 2.56 | 0.643[a] |
| Price | 183 | 55 | 80 | 21 | 22 | 5 | 10.67 | **0.031**[a] |
| Brand | 195 | 63 | 68 | 23 | 27 | 14 | 9.97 | **0.042** |
| Origin | 163 | 46 | 56 | 26 | 23 | 12 | 6.07 | 0.195 |
| Taste (knows pet likes it) | 276 | 87 | 98 | 37 | 36 | 18 | 16.09 | **0.003** |
| Veterinary recommendation | 219 | 66 | 86 | 28 | 28 | 11 | 2.91 | 0.568 |
| Pet store associate recommendation | 127 | 37 | 58 | 19 | 10 | 3 | 8.75 | 0.064[a] |
| Online recommendation | 118 | 37 | 63 | 12 | 6 | 0 | 28.20 | **<0.001**[a] |
| Friend/family/co-worker recommendation | 175 | 59 | 76 | 22 | 16 | 2 | 21.16 | **<0.001**[a] |
| **Method of deciding on amount to feed**[*] | | | | | | | | |
| Follows recommended daily feeding amount on package label | 99 | 37 | 37 | 12 | 10 | 3 | 6.66 | 0.175[a] |
| Veterinary advice | 92 | 19 | 42 | 22 | 7 | 2 | 17.54 | **0.002**[a] |

*(Continued)*

**Table 4.** (Continued)

| Variable | Total (n = 337) | Caregiver age | | | | | P-Chi² | P-value |
|---|---|---|---|---|---|---|---|---|
| | | 18–25 (n = 96) | 26–35 (n = 134) | 36–45 (n = 47) | 46–59 (n = 39) | 60+ (n = 20) | | |
| Own research | 60 | 25 | 25 | 5 | 1 | 4 | 12.39 | **0.008[a]** |
| Monitors pet's BCS | 121 | 45 | 38 | 16 | 14 | 8 | 8.54 | 0.072[a] |
| Based on pet's recent activity level | 52 | 20 | 20 | 8 | 1 | 3 | 7.20 | 0.080[a] |
| Until pet seems satisfied | 56 | 14 | 28 | 5 | 7 | 2 | 3.94 | 0.467[a] |
| No specific strategy | 63 | 16 | 22 | 6 | 12 | 7 | 9.02 | 0.070[a] |
| Other | 14 | 2 | 4 | 1 | 5 | 2 | 11.02 | **0.034[a]** |
| **Method of delivery (very often/always)[*]** | | | | | | | | |
| Traditional food bowl | 91 | 24 | 45 | 12 | 8 | 2 | 6.94 | 0.150[a] |
| Interactive puzzle/slow feeder/food dispense ball | 66 | 14 | 44 | 6 | 2 | 0 | 27.84 | **<0.001[a]** |
| Electronic/smart feeder | 34 | 9 | 19 | 5 | 1 | 0 | 7.20 | 0.135[a] |
| Hand feeding | 221 | 81 | 72 | 21 | 31 | 16 | 37.73 | **<0.001** |
| Stuffed toys | 39 | 12 | 19 | 7 | 1 | 0 | 7.17 | 0.105[a] |
| Other | 26 | 5 | 15 | 3 | 1 | 2 | 4.83 | 0.321[a] |

Numbers of caregivers (rows) may not add up to total due to exclusion of the 'prefer not to answer' caregiver age category (n = 1) from analyses. Bolded values indicate significance (p<0.05).

[a]Fisher's exact test

[*]Respondents could select more than one response option

(Table 3). For treats, most (121/337, 36%) caregivers indicated that they monitor their cat's BCS to decide on the amount to feed (Table 4). While there were no significant differences among how respondents decide on the amount to feed their cat for the primary diet by caregiver age group, there were significant (p<0.034) differences in the proportion of cat caregivers who use veterinary advice, their own research, and an 'other' strategy to decide on the amount to feed their cat for treats by caregiver age group (Table 4). Generally, middle-aged (36–45) caregivers reported relying more on veterinary advice, while caregivers in younger age groups reported relying on their own research. Furthermore, there were significant (p<0.019) differences in the proportion of cat caregivers who use veterinary advice and have no specific strategy to decide on the amount to feed their cat for the primary diet versus treats (p<0.05) (Fig 1). Specifically, more caregivers reported relying on veterinary advice for decisions surrounding feeding amounts for their cat's primary diet compared to treats, while more caregivers reported having no specific strategy on feeding amounts for treats compared to the primary diet.

**Monitoring treat intake.** The majority (224/337, 66%) of cat caregivers reported that they do monitor their pet's treat intake, while 21% (72/337) sometimes monitor their pet's treat intake, and 12% (41/337) do not monitor their pet's treat intake. Caregivers were significantly more likely to monitor their cat's treat intake if they considered their cat a 1/5 (very thin) (OR = 10.05, p<0.001), or 3/5 (normal/ideal) (OR = 5.451, p<0.001) on the BCS chart (Table 5). Respondents who provide treats that are pre-portioned when purchased (OR = 5.30, p = 0.003) or use an 'other' method of measurement for treats (OR = 5.55, p = 0.002) were also significantly more likely to monitor their pet's treat intake (Table 5).

On average, caregivers estimated that treats account for a median of 15% (range of 1%-99%) of their cat's total diet, based on estimated quantity. Caregivers who consider treats part of their cat's normal diet (median = 46%, range of 1%-99%) or who consider some treats part of their pet's normal diet and some extra (median = 38%, range of 1%-99%) estimated treats to

**Table 5. Final multivariable logistic regression models exploring the association between (1) whether cat caregivers monitor pet's treat intake (yes/no outcome) and method of which they measure their pet's treats (explanatory), and (2) caregiver-reported BCS as overweight/obese (yes/no outcome) and frequency of feeding different types of treats (explanatory) by cat caregivers (n = 337).**

| Variable | Odds Ratio | 95% CI | P-value |
|---|---|---|---|
| **Model 1[a]** | | | **<0.001** |
| **Cat age (ref = <1 year)** | | | **<0.001** |
| 1–3 years | 0.19 | 0.05–0.72 | **0.015** |
| 4–6 years | 0.52 | 0.12–2.25 | 0.379 |
| 7+ years | 0.34 | 0.08–1.40 | 0.135 |
| **Cat sex (ref = male intact)** | | | **<0.001** |
| Male neutered | 3.78 | 1.46–9.81 | **0.006** |
| Female intact | 1.07 | 0.33–3.41 | 0.915 |
| Female spayed | 1.91 | 0.70–5.20 | 0.206 |
| **Caregiver-reported BCS rating (ref = 5/obese)** | | | **<0.001** |
| 1/very thin | 10.05 | 3.38–29.87 | **<0.001** |
| 2/underweight | 2.89 | 0.93–8.98 | 0.067 |
| 3/normal/ideal | 5.51 | 2.21–13.70 | **<0.001** |
| 4/overweight | 2.11 | 0.73–6.08 | 0.168 |
| **Method of measurement for treats*** | | | **<0.001** |
| Servings are pre-portioned | 5.30 | 1.73–16.20 | **0.003** |
| Do not measure | 0.06 | 0.03–0.16 | **<0.001** |
| Other | 5.55 | 1.89–16.27 | **0.002** |
| **Model 2[b]** | | | **<0.001** |
| **Caregiver age (ref = 18–25)** | | | **<0.001** |
| 26–35 | 1.00 | 0.48–2.09 | 0.996 |
| 36–45 | 0.31 | 0.10–0.92 | **0.034** |
| 46–59 | 0.40 | 0.14–1.13 | 0.083 |
| 60+ | 0.34 | 0.08–1.44 | 0.142 |
| **Caregiver gender (ref = man)** | | | **<0.001** |
| Woman | 3.38 | 1.35–8.42 | **0.009** |
| Non-binary | 1.79 | 0.22–14.36 | 0.583 |
| **Cat age (ref = <1 year)** | | | **<0.001** |
| 1–3 years | 1.09 | 0.25–4.81 | 0.910 |
| 4–6 years | 4.33 | 0.87–21.44 | 0.073 |
| 7+ years | 9.37 | 1.86–47.12 | **0.007** |
| **Feeding frequencies for treats*** | | | **<0.001** |
| Training treats, a few times a week | 0.39 | 0.17–0.88 | **0.023** |
| Jerky, daily | 5.12 | 1.42–18.43 | **0.013** |
| Jerky, never | 4.02 | 1.10–14.73 | **0.036** |
| Bones, a few times a week | 14.09 | 3.62–54.88 | **<0.001** |
| Bones, weekly | 10.45 | 3.04–35.88 | **<0.001** |
| Dental treats, a few times a week | 3.48 | 1.67–7.22 | **0.001** |

(*Continued*)

**Table 5.** (Continued)

| Variable | | Odds Ratio | 95% CI | P-value |
|---|---|---|---|---|
| | Table scraps, daily | 4.22 | 1.21–14.69 | **0.024** |

Bolded values indicate significance (p<0.05)

*These variables contain categories that were explored as individual variables and as such, the referent category entails respondents who did not select these categories.

P-values presented in the row of the variable represent the results of the likelihood ratio test (LRT). In the case where categories were explored as individual variables, the highest p-value is provided. The overall model p-values are presented in the row of each model title.

[a]The variables caregiver age, gender, education, cat age, sex, and caregiver-reported BCS rating were entered into the model as suspected confounders. The variable caregiver age was identified as a confounding variable and retained in the model.

[b]The variables caregiver age, gender, education, cat age, and sex were entered into the model as suspected confounders. The variables caregiver education, and sex of cat were identified as confounding variables and retained in the model.

account for significantly more of their cat's total diet, compared to caregivers who considered all treats additional to their cat's normal diet (median = 9%, range of 1%-93%) ($\chi^2$ = 116.7, p<0.001). Caregivers who were 'unsure' estimated treats to account for a median of 10% (range of 10%-10%) of their cat's diet.

Moreover, the amount of treats estimated by caregivers to account for their cat's total diet varied by caregivers' type of relationship ($\chi^2$ = 210.7, p<0.001) and level of attachment with their cat ($\chi^2$ = 140.9, p<0.001) (Table 6). Specifically, caregivers who considered their cat a companion estimated treats to account for a significantly less amount of their pet's diet compared to caregivers who considered their cat a teammate/partner (p = 0.025). Further,

**Table 6. Median percentage of diet that comprised of treats, based on total estimated quantity, compared to the type of relationship and level of attachment reported by cat caregivers (n = 337).**

| Variable | n | Median percent of diet comprised of treats, based on estimated quantity | Range |
|---|---|---|---|
| Type of relationship with cat | | | |
| Like a child | 117 | 15% | 98 |
| Part of my family | 170 | 15% | 94 |
| Teammate/partner | 9 | 51% | 54 |
| Companion | 36 | 9% | 88 |
| Acquaintance | 3 | 10% | 56 |
| Prefer not to answer | 1 | 15% | 0 |
| Level of attachment | | | |
| 1 | 0 | - | - |
| 2 | 0 | - | - |
| 3 | 1 | 2% | 0 |
| 4 | 4 | 57% | 58 |
| 5 | 14 | 52% | 71 |
| 6 | 12 | 45% | 74 |
| 7 | 12 | 29% | 79 |
| 8 | 24 | 34% | 94 |
| 9 | 72 | 25% | 98 |
| 10 | 136 | 10% | 98 |

caregivers who reported their level of attachment with their cat as a 10/10 estimated treats to account for a significantly less amount of their pet's diet compared to caregivers who rated their level of attachment as a 5/10 (p = 0.003), 6/10 (p = 0.013), 8/10 (p = <0.001), and 9/10 (p<0.001).

**Factors influencing treat selection and provision.** Taste (knowing their pet likes it) was the most common (276/337, 82%) factor reported by caregivers to influence what treats they feed their cat, followed by the main ingredient (245/337, 73%) (Table 4). More than 50% of respondents also indicated the complete ingredient composition, size, health claims, price, brand, veterinary recommendation, and friend/family/co-worker recommendation as influencing factors for deciding on what treats to feed their cat (Table 4). The influencing factors price, brand, online recommendations, and friend/family/co-worker recommendations differed significantly (p<0.042) by caregiver age group; generally higher proportions of caregivers in younger age groups indicated they would be influenced by these factors compared to caregivers in older age groups (Table 4).

Caregivers expressed a variety of reasons for providing treats to their cat, though the pet-caregiver relationship was a common theme among these results. Knowing their cat enjoys treats was the most reported motivating factor for caregivers to provide treats (199/337, 59%) (Table 7), and the most common reason respondents reported frequently (always/very often) providing treats (185/337, 55%) (Fig 2). More than half of caregivers also reported frequently providing treats to show love to their pet (178/337, 53%) (Fig 2), and nearly 50% of respondents were motivated to feed treats to enrich their pet's day (166/337, 49%), because it makes themselves as the caregiver happy (158/337, 47%), and because it strengthens the bond with their pet (153/337, 45%) (Table 7).

## Different types of treats

Cookies (134/337, 40%), soft and chewy/training treats (116/337, 35%), and dental treats (114/337, 34%) were the most popular types of treats fed either daily or a few times a week by cat caregivers (Table 8). Significantly more cats considered very thin (1/5 BCS) by their caregiver were reported to receive human food frequently (daily or a few times a week), compared to cats considered underweight (2/5 BCS), normal/ideal (3/5 BCS), and overweight (4/5 BCS), but not compared to cats considered obese (5/5 BCS) (p<0.003). Cats fed table scraps (OR = 4.22, p = 0.024) or jerky (OR = 5.12, p = 0.013) daily had significantly higher odds of

**Table 7. Reported factors motivating cat caregivers to feed treats (n = 337).**

| Factors that motivate cat caregiver to feed treats* | n | % |
|---|---:|---:|
| Makes pet happy | 199 | 59.1 |
| To enrich pet's day | 166 | 49.3 |
| Makes caregiver happy | 158 | 46.9 |
| Strengthens bond with pet | 153 | 45.4 |
| Reinforcing desired behaviour | 141 | 41.8 |
| Pet anticipates treats as part of their routine | 120 | 35.6 |
| When it appears pet asks for treats | 107 | 31.8 |
| Nutritional/health benefits | 81 | 24.0 |
| To keep pet busy/occupied | 51 | 15.1 |
| Other | 13 | 3.9 |
| Unsure | 0 | 0.0 |

*Respondents could select more than one response option

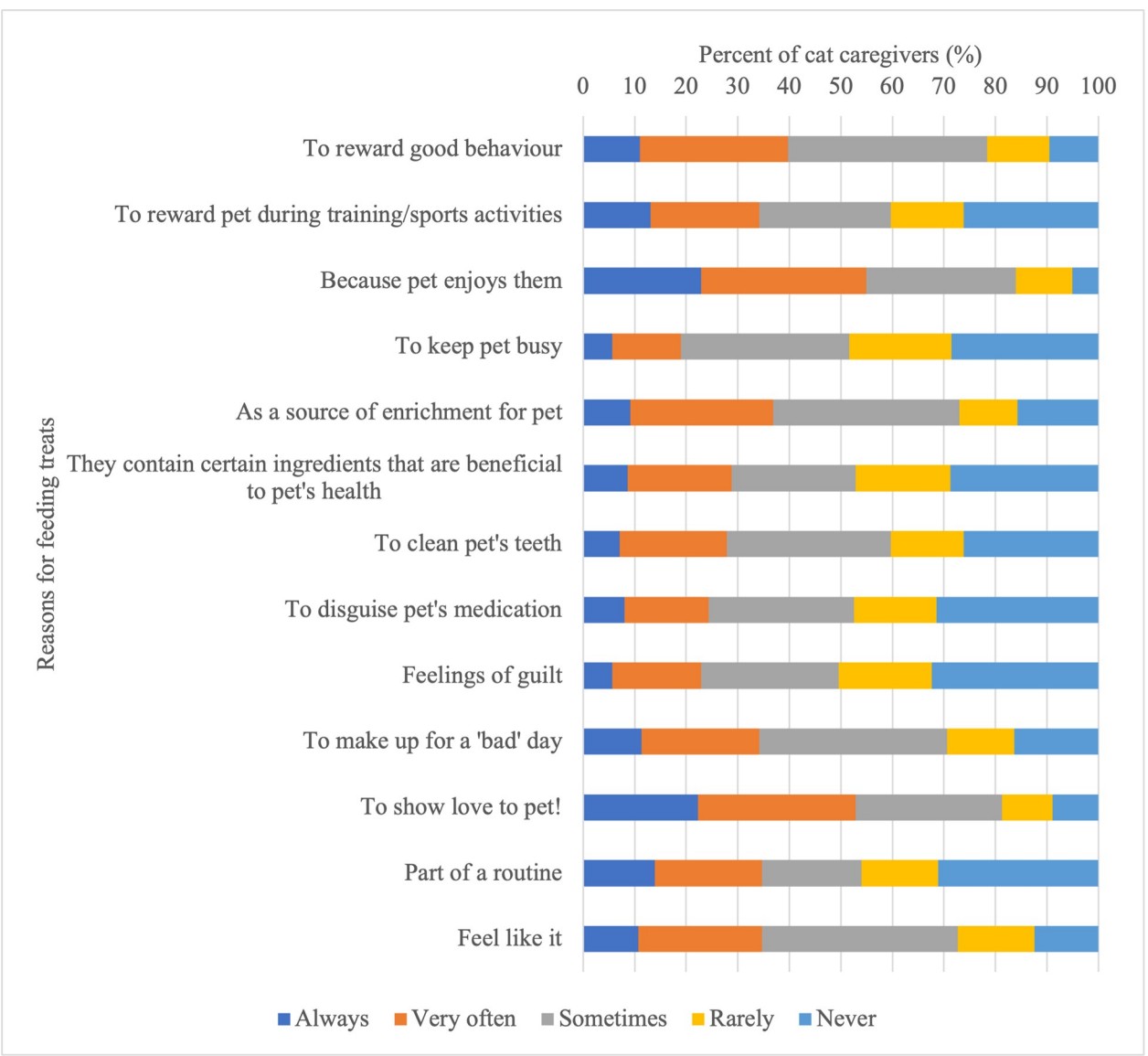

**Fig 2. Frequency of reasons for feeding treats as reported by cat caregivers (n = 337).**

being perceived as overweight or obese by their caregiver, though cats fed training treats a few times a week (OR = 0.39, p = 0.023) were significantly less likely to be perceived as overweight or obese (Table 5).

Most cat caregivers reported feeding two different types (116/337, 34%) and two different brands (119/337, 35%) of treats, though regarding different flavour varieties of treats most respondents reported feeding three to five (126/337, 37%) (Fig 3). Introducing variety was the most reported motivating factor to feed different flavour varieties (150/337, 45%) and brands (103/337, 31%) of treats, and most (119/337, 35%) of caregivers were motivating to feed different types of treats because each treat serves a different purpose (Fig 3). Many caregivers were also motivated to feed different flavour varieties (107/337, 32%) and different types (110/337, 33%) of treats to enrich their pet's life, though 20–30% of cat caregivers indicated that they only feed one type, brand, or flavour variety of treat to their cat (Fig 3).

**Table 8. Reported type and frequency of treats fed by cat caregivers.**

| Types of treats fed to pet frequently (daily or a few times a week)* | Total (n = 337) | Caregiver-reported rating of pet's BCS on 5-point scale, n (%) | | | | |
|---|---|---|---|---|---|---|
| | | 1 (n = 47) | 2 (n = 38) | 3 (n = 148) | 4 (n = 46) | 5 (n = 52) |
| Cookies | 134 (39.8) | 24 (51.1) | 14 (36.8) | 55 (37.2) | 9 (19.6) | 32 (61.5) |
| Soft and chewy/training | 116 (35.1) | 25 (53.2) | 17 (44.7) | 43 (29.1) | 7 (15.2) | 24 (46.2) |
| Jerky | 64 (19.3) | 22 (44.8) | 2 (5.3) | 16 (10.8) | 2 (4.4) | 22 (42.3) |
| Chews | 53 (16.0) | 13 (27.7) | 2 (5.3) | 21 (14.2) | 0 (0.0) | 17 (32.7) |
| Bones | 54 (16.3) | 17 (36.2) | 6 (15.8) | 12 (8.1) | 0 (0.0) | 19 (36.5) |
| Dental treats | 114 (34.4) | 15 (31.9) | 8 (21.1) | 47 (31.8) | 18 (39.1) | 26 (50.0) |
| Human food | 78 (23.6) | 23 (48.9) | 7 (18.4) | 24 (16.2) | 8 (17.4) | 16 (30.8) |
| Table scraps | 66 (19.9) | 14 (29.8) | 4 (10.5) | 19 (12.8) | 6 (13.0) | 23 (44.2) |

Caregiver-reported rating of pet's BCS on 5-point scale (rows) may not add up to total due to exclusion of the 'unsure' category (n = 6) from analyses.

*Respondents could select more than one response option

## Discussion

Previous studies investigating feline feeding habits have mainly focused on aspects relating to the primary diet, such as diet type and feeding practices [14, 31, 41]. While some studies have explored dog owners' attitudes towards treats [32, 33], research has yet to focus on treat feeding with cats specifically. This study provides new insight to cat caregivers' perceptions, motivations, and behaviours related to treat feeding, thus contributing to the current gap in the literature.

A range of perspectives were shared by respondents on the meaning of the term 'treat' with more than half considering the term strictly through a nutritional lens, either relating to food in general or specifically as commercial pet treats. This nutritional perspective is shared among dog caregivers, with even lower proportions including factors external to nutrition in their definition [32]. These different perspectives of the term 'treat' amongst caregivers could have practical implications and highlight the need for clear and consistent communications for cat caregivers regarding treats to effectively promote companion animal health and well-being. For instance, caregivers who view treats in a commercial context exclusively may be more inclined to purchase and offer only specific types of commercial pet treats, potentially affecting the variety and balance of nutrients in their cat's diet which could contribute to nutrient deficiencies or toxicities [42, 43]. On the other hand, caregivers who view treats more broadly as any food they give their cat that they enjoy may offer unsuitable human foods which aside from causing nutrient imbalance, could have specific negative health implications (e.g., *Allium* toxicosis from ingestion of onions) [44]. Further, caregivers who consider treats solely in a nutritional context may have a limited view on how they can interact with their cat, potentially over-relying on food, which could lead to overfeeding and contribute to obesity. Future research could investigate how these differing perspectives of the term 'treat' may impact caregiver feeding decisions and behaviours, and feline health outcomes.

Caregivers also expressed mixed views on how they consider treats in relation to their cat's normal diet. Most respondents considered all, or at least some treats to be part of their pet's normal diet, and these caregivers estimated treats to account for a significantly greater quantity of their pet's diet, compared to caregivers who considered treats exclusively additional to their pet's normal diet. According to the World Small Animal Veterinary Association (WSAVA) Global Nutrition Committee, treats should not account for more than 10% of a pet's total calories [43]. Our results suggest that most cat caregivers may be feeding a greater proportion of treats than would be recommended. Though diet history information was not collected in the

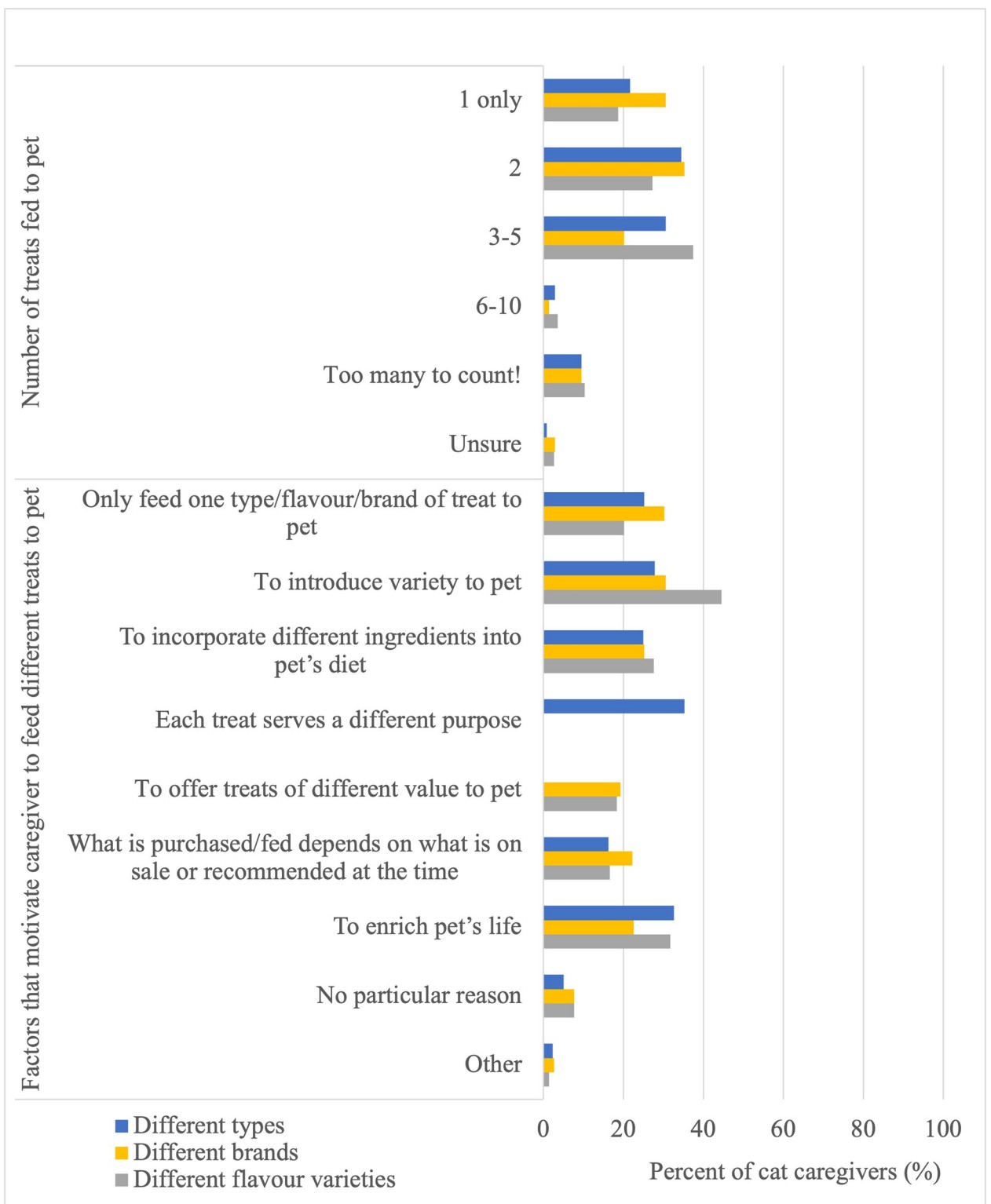

**Fig 3. Number of different types, brands, and flavour varieties of treats fed to their pet, and factors that motivate them to feed different types, brands, and flavour varieties of treats as reported by cat caregivers (n = 337).**

present study to verify feeding proportions and calculate the proportion of calories from treats and the cat's primary diet, overconsumption of treats can lead to obesity and a variety of co-morbidities that may negatively impact feline health and well-being [13, 17, 45]. Additionally, caregivers who feed excess treats may inadvertently provide an unbalanced diet, as treats often lack the requirements of complete and balanced nutrition and are therefore according to the Association of American Feed Control Officials (AAFCO), labeled for intermittent feeding only [42, 43]. Subsequent studies may more specifically investigate which treats cat caregivers may consider part of their pet's diet, and which treats they may consider extra, to further the knowledge surrounding cat caregiver decisions involving treats.

Most caregivers in our study reported monitoring their pet's treat intake, though most measured treats using an eyeball estimate and fewer than 30% utilized veterinary advice on feeding amounts for treats. Previous research has shown that caregivers are less likely to rely on veterinary advice for feeding amounts compared to decisions on what to feed [46]. Interestingly, our study found significant differences in the use of veterinary advice for treat feeding among different age groups of caregivers. Younger caregivers were more likely to rely on their own research and online recommendations, which can produce inconsistent and incomplete results, resulting in inappropriate practices [47–49]. Additional veterinary outreach to younger caregivers may have benefit for cat health in promoting reliable sources of information from which caregivers can make informed decisions not only about what to feed, but also how much to feed. Though previous research suggests that cat caregivers are more likely to recognize the benefits of veterinary nutrition care for their feline companions compared to dog caregivers, they are also less likely to take their cats to the veterinarian more than once per year [46]. Neglecting such visits can exacerbate existing challenges surrounding time constraints in veterinary appointments, which have been identified as a communication obstacle within the profession [50]. In addition, obtaining a comprehensive dietary history from caregivers can be a complex process, requiring specific communication strategies [51]. Our findings suggest that most caregivers feed multiple types, brands, and flavours of treats, often motivated by the desire to introduce variety. Given the importance of nutrition in overall companion animal care, it is crucial to encourage cat caregivers to schedule regular veterinary visits. By doing so, veterinarians are presented with valuable opportunities to obtain accurate diet history and offer tailored nutritional guidance, such as appropriate amounts for treats, while also being able to monitor other important indicators of their patient's health and well-being.

Our results highlight the role of the human-animal bond in treat feeding amongst cat caregivers. Perceiving treats make their pet happy, enrich their pet's day, and strengthen the bond with their pet were among the most reported motivating factors to feed treats by respondents. The concept of using food to convey love and affection has been previously suggested in companion animal research [22, 32, 34]. More than half of the caregivers in our study considered their cat a part of their family, and 35% considered their cat like a child. Moreover, nearly all caregivers expressed a high level of attachment with their cat. These results align with existing literature highlighting the perceived strength of the human-animal bond which can emulate human relationships [52–58]. A significantly greater proportion of caregivers in younger age categories considered their cat like a child, while caregivers in older age categories were more likely to view their cat as a family member. Relationship dynamics should be considered when discussing aspects surrounding feeding with caregivers, especially since our findings suggest that levels of attachment and types of relationships could influence the amount of treats caregivers provide to their pet. Interestingly, caregivers in our study who rated their attachment level with their cat as 10/10 reported that treats comprised only a median average of 10% of their cat's diet, aligning with WSAVA guidelines [43]. In contrast, caregivers with slightly lower but still high levels of attachment (i.e., 8/10, 9/10) tended to report higher proportions.

Future research is needed to determine whether caregivers most attached to their cats are more likely to adhere to feeding guidelines or if their awareness of such recommendations leads to a higher likelihood of reporting these practices (i.e., social desirability bias). Given that owners rated their perceived level of happiness upon providing a treat to their cat significantly higher than their cat's perceived happiness, our findings reinforce the idea that treat feeding amongst cat caregivers is not just about providing nutrition but is also an important aspect of the human-animal bond.

Cat caregivers who have a close relationship with their feline companion should be particularly careful when it comes to feeding amounts for treats. Previous research suggests that a close relationship between cat and owner is a risk factor for feline obesity, with tendencies to overfeed them [22]. This overfeeding can happen when caregivers use food as treats or give in to apparent begging [22]. Our study found that cats who were fed table scraps daily were more likely to be perceived as overweight or obese by their caregivers. On the other hand, results by Kienzle and Bergler [22] further revealed that caregivers with cats considered a normal weight were more likely to reward their cats with extra play time. Providing training treats a few times a week was the only treat type found to be protective of feline overweight and obesity in our results. While our survey did not explore specific details on the criteria for providing training treats, these findings present an opportunity for caregivers to continue to utilize treats to interact and bond with their cat, while minimizing excess calories by promoting physical activity alongside treat provision.

Our results demonstrate the importance caregivers perceive their cats enjoy treats and the role that treats play in showing love to them. Caregivers who are motivated to enrich their cat's day can do so by incorporating enrichment feeding methods such as food puzzles, which have been shown to have positive effects on cat behaviour, including reduced anxiety, fear, and aggression [59]. However, only a minority of caregivers reported using these methods frequently in our study and in other publications [60], highlighting the need for greater education and awareness about enrichment feeding practices. Veterinarians are well-placed to provide information and education about enrichment feeding methods and can encourage caregivers to implement them as part of their cat's feeding routine. By doing so, caregivers can not only experience greater satisfaction in their caregiving role and human-animal relationship, but also contribute to their cat's overall well-being. Due to their preparatory nature, enrichment feeders may further support caregivers in managing treat provision to appropriate quantities since they are often designed to be used with specific portion sizes. Additionally, these feeding tools present an opportunity for veterinarians to provide guidelines in the form of recipes that caregivers could use with them, to ensure cats are receiving appropriate nutrition. Information about enrichment feeding methods for caregivers could be the form of supplementary 'take home' material provided either at the time of a veterinary appointment or after.

The respondent population had a greater proportion of caregivers identifying as women, consistent with previous research involving companion animals [32, 33, 46, 61, 62]. This study was limited by the fact that we used a convenience sample of caregivers, and the findings must be interpreted with recognition of inherent bias associated with this methodology. The sampling strategy employed allowed for self-selection into the study which likely introduces bias with respect to the nature of participants. The intention of employing this sampling strategy was to obtain a large sample from the general cat caregiver population, though it is likely that cat caregivers especially invested in their cat's nutrition and well-being were most motivated to participate. As such, our results could reflect a subpopulation of caregivers who are highly perceptive of their cat's health and wellness, resulting in particular feeding practices. This could be one reason why most caregivers reported their cat to be of ideal (3/5) body condition and fewer respondents reported their cat to be overweight or obese compared to other recent

results [18, 63]. The majority of cats described were relatively young, between 1–3 years of age, which could also explain the prevalence of obesity observed in our study [25], The fact that caregivers self-reported their cat's BCS in our study is another limitation and could also have had an effect on the results, as previous research has demonstrated that caregivers tend to underestimate their pet's body condition [23, 64, 65]. For example, this could be one reason as to why the multivariable logistic regression model produced conflicting results whereby feeding jerky either daily or never were both predictive of perceived overweight/obesity. It is also possible that responses were affected by the non-random presentation of survey items, and social desirability bias could have influenced accuracy in reporting by caregivers. Though it was stressed to participants that our questionnaire was anonymous and efforts to maintain confidentiality of personally identifying data were outlined to best support caregivers in completing the questionnaire truthfully. Further, not all terms presented in the questionnaire were defined and participants could have interpreted them differently, which could have resulted in misclassification bias. Since the questionnaire was presented online, it was not possible to establish the number of potential cat caregivers who chose not to participate (non-respondents). This further impedes the opportunity to evaluate how representative the sample is of the general cat caregiver population. Finally, findings presented in this study solely reflect the perceptions of cat caregivers and must be interpreted as such.

## Conclusion

This study provides valuable insights to the perceptions, motivations, and behaviours of cat caregivers surrounding treat feeding, which is an area that has received little attention in previous research. These results have practical implications for veterinarians and pet food manufacturers, as well as for cat caregivers looking to make informed decisions about their pet's diet and treat intake. Caregivers hold varying interpretations of the term 'treat' and how they relate to their cat's regular diet, and these perceptions seem to impact the quantity of treats provided. The results suggest that the human-animal bond is a key factor that should be taken into consideration when designing interventions aimed at promoting healthy feeding habits in cats. An understanding of the effects of treat feeding on feline nutrition, behaviour, and overall well-being is critical to informed decision making and ultimately, improved feline health outcomes. Additional research around treat feeding for cats can further contribute our knowledge surrounding the impact of this practice on feline health and facilitate the development of effective guidelines for optimal feeding practices.

## Supporting information

**S1 Appendix. Survey questions exploring caregiver perceptions and behaviours about treats.**
(PDF)

## Author Contributions

**Conceptualization:** Shelby A. Nielson, Deep K. Khosa.

**Formal analysis:** Shelby A. Nielson.

**Funding acquisition:** Deep K. Khosa.

**Investigation:** Shelby A. Nielson.

**Methodology:** Shelby A. Nielson, Deep K. Khosa.

**Project administration:** Shelby A. Nielson, Deep K. Khosa.

**Supervision:** Deep K. Khosa, Adronie Verbrugghe, Katie M. Clow.

**Writing – original draft:** Shelby A. Nielson.

**Writing – review & editing:** Deep K. Khosa, Adronie Verbrugghe, Katie M. Clow.

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
