## [Decision Letter · Decision Letter 0]

26 Oct 2023

PONE-D-23-20840Cat caregivers’ perceptions, motivations, and behaviours for feeding treats: a cross sectional studyPLOS ONE

Dear Dr. Nielson,

Thank you for submitting your manuscript to PLOS ONE. After careful consideration, we feel that it has merit but does not fully meet PLOS ONE’s publication criteria as it currently stands. Therefore, we invite you to submit a revised version of the manuscript that addresses the points raised during the review process.

We look forward to receiving your revised manuscript.

Kind regards,

Harvie P. Portugaliza, D.V.M., Ph.D.

Academic Editor

PLOS ONE

Journal Requirements:

"A.V. is the Royal Canin Veterinary Diets Endowed Chair in Canine and Feline Clinical Nutrition and declares that they serve on industry advisory boards and received honoraria and research funding from various pet food manufacturers and ingredient suppliers."

4. We note that Figures S1 Appendix in your submission contain copyrighted images. All PLOS content is published under the Creative Commons Attribution License (CC BY 4.0), which means that the manuscript, images, and Supporting Information files will be freely available online, and any third party is permitted to access, download, copy, distribute, and use these materials in any way, even commercially, with proper attribution. For more information, see our copyright guidelines: http://journals.plos.org/plosone/s/licenses-and-copyright.

a. You may seek permission from the original copyright holder of Figurs S1 Appendix to publish the content specifically under the CC BY 4.0 license. 

Reviewers' comments:

Reviewer's Responses to Questions

**Comments to the Author**

1. Is the manuscript technically sound, and do the data support the conclusions?

Reviewer #1: Yes

Reviewer #2: Yes

2. Has the statistical analysis been performed appropriately and rigorously? 

Reviewer #1: Yes

Reviewer #2: Yes

3. Have the authors made all data underlying the findings in their manuscript fully available?

Reviewer #1: Yes

Reviewer #2: Yes

4. Is the manuscript presented in an intelligible fashion and written in standard English?

Reviewer #1: Yes

Reviewer #2: Yes

5. Review Comments to the Author

Reviewer #1: This article is indispensable in addressing the questions on the role of cat owners and caregivers toward feline nutrition. It contributes knowledge for a comprehensive understanding of the factors surrounding feline nutrition, particularly human behavior. The authors gave adequate background information on the prevalence of feline obesity, its comorbidities, and the role of treat feeding among other factors. The researchers attempted to address the question relating to cat caregivers’ behaviors, perceptions, and decisions in providing treats to their cats by conducting a cross-sectional survey and analyzing the data to identify significant factors. The authors found out that caregivers perceive treat as anything that their pet enjoys, including food items (i.e., commercial pet treats), and are considered part of the normal feline diet. Most of the caregivers selected the type of treats primarily based on the taste (knowing their pet likes it). Friends, family, coworkers, and online recommendations; as well as price and brand significantly influence younger caregivers in selecting the type of treats. Decisions on the amount of treats given to their pets were made mainly based on the body condition score (BCS). Middle-aged caregivers rely more on veterinary advice, while caregivers in younger age groups rely on their own research. In general, the article provides valuable insights into the perceptions, motivations, and behaviors of cat caregivers surrounding treat feeding, which is an area worth exploring. I recommend publishing the article.

The main strength of the article is that it addresses the questions on social aspects relating to feline nutrition and to the prevalence of feline obesity by providing clear information on the behaviors, perceptions, and motivations of cat caregivers. Importantly, the article conforms to the STROBE checklist.

Weaknesses of the article are well stated in the discussion section such as potential bias due to the use of convenience sampling and self-selection in the study, and the results may only reflect a subpopulation of caregivers. Nevertheless, the authors stipulated it to give the readers a precaution for interpreting the results. There is also a lack of evidence for internal consistency for Likert-scale questions in the questionnaire used in the study. I suggest further revision of the article as there are minor typographical errors and needed improvements and clarifications.

For the improvement of the article, the authors are encouraged to resolve the following minor issues:

1. In line 33 the authors claim the model is predictive, however, there are no mention of meeting assumption requirements. Hence it is suggested that the authors perform assumption checks (i.e. multicollinearity, VIF, etc.) and mentioned in the data analysis section. The overall model p-value should be reported in the multivariable tables. To further know how well the logistic regression model classifies the outcome, predicted probabilities can be used against the dependent variable to make an ROC curve and the area under the curve can be reported (Homer and Lemeshow, 2000).

2. The authors may also mention estimates of the prevalence of obesity in domestic cats in Canada in the second paragraph of the Introduction section.

3. Typo error in line 64 “white”.

4. In Line 75-78 the authors described “caregiver”. The authors should clarify if this is different from cat owners especially that in Table 1 other people in household are considered.

5. The authors should mention the questionnaire assessment and its results in data analysis section, e.g. Cronbach’s alpha test for Likert-scale questions

6. The authors should clarify lines 147-148 where it mentions differences in means when in fact the preceding sentences mention categorical variables.

7. The authors should present the mean or median and SD or IQR for continuous variables in Table 1 in line 201.

8. In Table 1, the authors only considered the “other people” living with the family. I think there is a need to consider for the family members because it’s not only the “other people” that contribute to pet’s care but the members of the family as well. The number of household members should have been relevant. The authors should clarify this.

9. In line 209, the n cats is only equal to 337 the same as the number of respondents. The authors should clarify if only 1 cat is considered per respondent. According to Table 1, the number of cats per household interviewed differed. For example, for 188 households there is 1 cat so 188 in total, 2 cats in 101 households, and 3 cats in 48 households so the total number of cats in the whole study should be = 534. The authors should give light to this discrepancy.

10. Typo error in Line 225 and 234, Table 3, instead of Table 4.

11. The author should state the abbreviation fully in Line 424 if first mentioned.

12. Typo error in Line 488.

References

Hosmer Jr., D.W and Lemeshow, S. (2000) Applied logistic regression. 2nd Edition, John Wiley & Sons, Inc., New York. http://dx.doi.org/10.1002/0471722146

Reviewer #2: Broad comments

A well-written manuscript illustrating valuable research aiming to understand perceptions and motivations surrounding provisioning treats to cats.

Specific comments

L41: remove ‘including’

L64: Perhaps a comment could be made that treat feeding is often perceived as a providing a reward and used during positive reinforcement training. There has been some research on this in dogs/cats, with the notion of improving their behavior and often used by owners with a stronger bond.

L164: stay consistent with terminology, it flows easier for the reader who may not be familiar with this analysis, if you describe the frequency of feeding treats as (explanatory dichotomous yes/no variables) like you explained the other model’s variables.

L173: revise: the ‘following’ categorical variables were included in this analysis due to their suspected role… : caregiver age … ,..

L217: how was this measured? Was this created for the survey? Did you use a certain scale? Lexington attachment to pets scales? Was this not used as a predictor to determine if it was related to their provision of treats (with the hypothesis that owners with a stronger bond are more likely to feed treats (regardless of BCS))? This would be interesting to explore as though we know cat obesity is such an immense welfare issue, and its often present within households that report a strong bond. I encourage exploring this relationship between attachment and BCS and their influence on provision of treats.

Table 5: were all partial f-tests for all these variables presented here significant? Would be useful to add these pvalues above the individual level pvalues (same row as the variable label).

- Method of measurement for treats – what was the referent?

- Feeding frequencies for treats – what was the referent?

6. PLOS authors have the option to publish the peer review history of their article (what does this mean?). If published, this will include your full peer review and any attached files.

Reviewer #1: No

Reviewer #2: **Yes: **Anastasia Stellato

---

## [Author Response · Author response to Decision Letter 0]

16 Nov 2023

Please see rebuttal letter that responds to each point raised by the academic editor and reviewed entitled, 'Response to Reviewers'.

---

## [Decision Letter · Decision Letter 1]

5 Dec 2023

Cat caregivers’ perceptions, motivations, and behaviours for feeding treats: a cross sectional study

PONE-D-23-20840R1

Dear Dr. Nielson,

We’re pleased to inform you that your manuscript has been judged scientifically suitable for publication and will be formally accepted for publication once it meets all outstanding technical requirements.

Kind regards,

Harvie P. Portugaliza, D.V.M., Ph.D.

Academic Editor

PLOS ONE

Additional Editor Comments (optional):

Reviewers' comments:

Reviewer's Responses to Questions

**Comments to the Author**

1. If the authors have adequately addressed your comments raised in a previous round of review and you feel that this manuscript is now acceptable for publication, you may indicate that here to bypass the “Comments to the Author” section, enter your conflict of interest statement in the “Confidential to Editor” section, and submit your "Accept" recommendation.

Reviewer #1: All comments have been addressed

Reviewer #2: All comments have been addressed

2. Is the manuscript technically sound, and do the data support the conclusions?

Reviewer #1: Yes

Reviewer #2: Yes

3. Has the statistical analysis been performed appropriately and rigorously? 

Reviewer #1: Yes

Reviewer #2: Yes

4. Have the authors made all data underlying the findings in their manuscript fully available?

Reviewer #1: Yes

Reviewer #2: Yes

5. Is the manuscript presented in an intelligible fashion and written in standard English?

Reviewer #1: Yes

Reviewer #2: Yes

6. Review Comments to the Author

Reviewer #1: Great. All the comments have been addressed. Optionally, the authors may remove the p-values in each variable row (LRT p-values). Since the p-values are all <0.001, they can just state it in one sentence footnote.

Reviewer #2: (No Response)

7. PLOS authors have the option to publish the peer review history of their article (what does this mean?). If published, this will include your full peer review and any attached files.

Reviewer #1: **Yes: **KENNY ORIEL ARANAS OLANA

Reviewer #2: **Yes: **Anastasia Stellato

---

## [Editor Report · Acceptance letter]

19 Dec 2023

PONE-D-23-20840R1 

PLOS ONE

Dear Dr. Nielson, 

I'm pleased to inform you that your manuscript has been deemed suitable for publication in PLOS ONE. Congratulations! Your manuscript is now being handed over to our production team.

Kind regards, 

on behalf of

Dr. Harvie P. Portugaliza 

Academic Editor

PLOS ONE